# A Low Dietary Quality Index in a Newly Diagnosed Inflammatory Bowel Disease Cohort: Results from a Case—Control Study

**DOI:** 10.3390/nu17060958

**Published:** 2025-03-10

**Authors:** Ravi Misra, Lovesh Dyall, Janet Kyle, Heather Clark, Jimmy Limdi, Rachel Cooney, Matthew Brookes, Edward Fogden, Sanjeev Pattni, Naveen Sharma, Tariq Iqbal, Pia Munkholm, Johan Burisch, Naila Arebi

**Affiliations:** 1IBD Department, St. Mark’s Hospital and Academic Institute, London NW10 7NS, UK; lovesh.dyall@nhs.net (L.D.); naila.arebi@imperial.ac.uk (N.A.); 2Department of Metabolism, Digestion and Reproduction, Imperial College London, London W12 0NN, UK; 3Postgraduate Education Group, Institute of Applied Health Sciences, University of Aberdeen, Aberdeen AB25 2ZD, UK; j.kyle@abdn.ac.uk (J.K.); h.clark@abdn.ac.uk (H.C.); 4Section of IBD, Northern Care Alliance NHS Foundation Trust, Manchester Academic Health Sciences, University of Manchester, Manchester BL9 7TD, UK; jimmy.limdi@nca.nhs.uk; 5University Hospitals Birmingham, Birmingham B12 2TH, UK; rachel.cooney@uhb.nhs.uk (R.C.); t.h.iqbal@bham.ac.uk (T.I.); 6Gastroenterology Unit, Royal Wolverhampton NHS Trust, London WV10 0QP, UK; m.j.brookes@bham.ac.uk; 7Sandwell and West Birmingham Hospitals, Birmingham B71 4HJ, UK; edward.fogden@nhs.net; 8University Leicester Hospitals, Leicester LE1 5WW, UK; sanjeev.s.pattni@uhl-tr.nhs.uk; 9Heartlands Hospital, Birmingham B9 5SS, UK; navsharma2411@gmail.com; 10Department of Gastroenterology, North Zealand University Hospital, 3600 Frederikssund, Denmark; pia_munkholm@mail.dk; 11Gastro Unit, Medical Section, Copenhagen University Hospital-Amager and Hvidovre, 2650 Hvidovre, Denmark; burisch@gmail.com; 12Copenhagen Center for Inflammatory Bowel Disease in Children, Adolescents and Adults, Copenhagen University Hospital-Amager and Hvidovre, 2650 Hvidovre, Denmark; 13Department of Clinical Medicine, Faculty of Health and Medical Sciences, University of Copenhagen, 2200 Copenhagen, Denmark

**Keywords:** inflammatory bowel disease, diet quality, dietary patterns

## Abstract

Background: Epidemiological evidence suggests a link between the risk of IBD and diet. Macro- and micro- nutrient intake, diet quality and dietary patterns may play a pivotal role in disease pathogenesis. We aimed to study the dietary intake of newly diagnosed IBD patients compared to non-IBD controls. Methods: A cohort of newly diagnosed IBD patients were invited to complete the Scottish Collaborative Group Food Frequency Questionnaire (SCGFFQ) at their first clinic visit. Controls were recruited from non-IBD ambulatory patients, university students, and healthcare workers. The SCGFFQ estimates habitual diet over a 3-month period. Component nutrient data were calculated based on previous validation studies, deriving nutrient data by comparison of the SCGFFQ to actual weighted food records. Data on age, gender, ethnicity, and disease phenotype were collected. The intake of macro- and micro-nutrients was expressed as mean and standard deviation and compared using the Kruskal–Wallis test. Dietary patterns were derived using principal component analysis. Differences in the dietary patterns for age, gender, and ethnicity were analysed by logistic regression analysis. The diet quality was compared to the dietary recommendation values (DRVs) and measured using the diet quality index. Results: We enrolled 160 IBD cases (114 UC and 46 CD) and 126 non-IBD controls, and in the study, with a median age across the groups of 40 years (IQR = 24) for UC, 34 years (IQR = 29) for CD, and 36 years (IQR = 24) for non-IBD controls. The diet quality indexes for both UC and CD were low compared to controls: 59.0% (SD 18.0) for UC, 46.0% (SD 17.7) for CD, and 63.2% (SD 17.1) controls. UC patients had excessive total energy consumption (>2500 kcal/day) compared to the DRVs. UC patients reported higher retinol, vitamin D, riboflavin, niacin, vitamin B6, vitamin B12, and panthanoic acid intake, consistent with a diet rich in animal products and low in fruit/vegetable intake. This is likely driven by higher consumption of dietary patterns 2 (rich in carbohydrates, refined sugar and low fibre) and 5 (refined sugar and saturated fat) in the UC cohort. Dietary pattern 1 (variety of food items and oily fish) was less likely to be consumed by the CD population. CD patients tended to have a lower overall intake of both macro- and micro-nutrients. Conclusions: The dietary patterns identified here are a proof of concept, and the next phase of the study would be to ideally monitor these patterns in a case–control cohort prospectively, and to further understand the mechanisms behind which dietary patterns influence IBD. Patients with newly diagnosed CD have low dietary quality and lower overall intake of macro- and micro-nutrients. This finding supports the role for dietetic attention early in newly diagnosed CD.

## 1. Introduction

Inflammatory bowel diseases (IBD), comprising Crohn’s disease (CD) and ulcerative colitis (UC), are a heterogenous group of chronic, inflammatory conditions affecting the gastrointestinal tract, characterised by a dysregulated immune response to the gut microbiota, with a multi-dimensional, and often negative impact on quality of life. An improved understanding of the triggers, and perpetuating factors of the immuno-inflammatory pathways will be pivotal to prevention and treatment.

Cumulative environmental exposures from conception, summarised as the “exposome”, have been shown, across several epidemiologic studies, to be associated with a risk of IBD [1,2]. Among these, breast milk consumption has been shown to have a protective effect against the risks of IBD in later life, whilst air pollution, particularly exposure to sulphur dioxide and nitric oxide, and early exposure to antibiotics in the first five years of life, may increase the risk of IBD [3,4,5].

Inflammatory bowel disease has become a global disease with accelerating incidence in newly industrialised countries whose societies have become more westernised. Although incidence is stabilising in Western countries, the burden remains high with the prevalence surpassing 0.3% [6]. In South Asia, there has been a rapid increase in IBD incidence and prevalence [7]. Diet remains a critical factor within the IBD exposome. Its importance is supported by epidemiological data on the lag of the increase in the exponential incidence of IBD in low–middle income countries compared with high income countries [8], where diets rich in saturated fats and sugars were adopted in the preceding decades [9]. The therapeutic benefits of diet lend strength to the hypothesis that an inflammatory diet perpetuates gut inflammation. Exclusive enteral nutrition (EEN) has consistently shown higher remission rates in (mainly paediatric) CD compared to placebo and steroid therapy [10,11]. Yet, the mechanism of action through which EEN induces remission in CD remains elusive [12].

The parallelism between westernisation, and particularly high dietary intakes of total fat (particularly animal fats, ω-6 polyunsaturated fatty acids (PUFA), and milk fats), refined sugars, meat and lower intakes of fruit and vegetables, implicates diet in the heightened risk of developing IBD [13,14]. However, this association remains inconsistent, underpinning the need for further study in this field [15].

Research directed at teasing out components of the diet, which are predisposed to non-communicable diseases, has revealed different dietary dimensions. These include dietary patterns, macro- and micro-nutrient intakes, and a comparison of diet quality compared to the recommended dietary guidelines [16].

Dietary research is fraught with challenges, not least recall bias, the manifold variables in dietary intake, the proportion of food intake relative to other dietary components, the potential for complex interactions between food groups, variable food metabolism among individuals, and inherent differences in food products [17,18].

Few studies have assessed dietary patterns in a recently diagnosed IBD population and have often been limited to one dietary dimension (e.g., diet quality, an individual micro- or macro- nutrient), with variable length of recall (24 h to 2 years), or through derivative analysis of data from large healthcare databases, or validated in one ethnic group [14,19,20].

To address this unmet need, we aimed to study the dietary pattern and nutrient intake of a newly diagnosed IBD cohort during the period preceding their diagnosis using a dietary questionnaire validated across multiple ethnic groups.

## 2. Methods

### 2.1. Study Participants

Newly diagnosed patients referred to specialist IBD UK centres were originally recruited in an inception cohort epidemiology IBD study [21]. Patients who completed their dietary data were included in this sub-study.

Controls were recruited from patients attending hospital for non-IBD-related medical conditions, healthcare workers, and university students. Controls were defined as any individual without any prior gastrointestinal diagnosis, including irritable bowel syndrome. The exclusion criteria included any self-reported diagnosis of obesity, any dietary restrictions, such as a low-salt diet for hypertension, calorie restrictive diets for weight loss, diabetes, and pregnancy. We aimed to match controls and cases for age, gender, and ethnicity.

### 2.2. Study Design

The Scottish Collaborative Group Food Frequency Questionnaire (SCGFFQ) was administered to patients at the first IBD clinic appointment in a treatment-naïve population. Demographic data on age, gender, ethnicity, smoking status, IBD sub-type, and Montreal classification were collected.

The SCGFFQ was used to capture habitual dietary intake over the preceding three months from the date of completion at the first hospital visits for IBD patients [22]. The questionnaire includes 175 of the most consumed items, divided into 19 sections. This tool has been previously validated against weighted food diaries to provide an estimate of the macro- and micro-nutrient intakes, in addition to deriving dietary patterns [23,24].

Clinical data and the SCGFFQ data collections were primarily paper records, with a change to telephone and online records during the COVID-19 pandemic, limiting data collection on height/weight and body mass index. To reduce the risks of recall bias, participants were encouraged to complete the SCGFFQ independently from the study researchers. SCGFFQs with five or more missing responses were excluded from analysis.

### 2.3. Data Management and Statistical Analysis

Data were securely collated using Microsoft Excel© and statistical analysis using SPSS version 29.0 (IBM SPSS Statistics© for Windows). Median and interquartile range (IQR) and mean with standard deviations (SD) were used for descriptive statistics for participant demographics, and nutrient intakes, respectively. The Kruskal–Wallis test was used to compare nutrient intakes across UC, CD, and controls. Statistical significance was defined as a *p*-value ≤ 0.05.

The Kruskal–Wallis test was used to allow the comparison of non-IBD, UC, and Crohn’s data, which were not normally distributed and non-parametric.

Macronutrient intake was compared with the UK government recommended dietary reference values (DRVs) [25].

The Diet Quality Index (DQI), whereby the relative diet of a study population is compared against a healthy diet for the average person in the UK, was used to compare intakes of the specific food groups (fruit, vegetables, fish, red, and processed meat products), along with the food nutrient content (fats, saturated fats, sugars and starch, fibre, and alcohol), generating a composite score calculated from the components of dietary quality, as assessed by “adequacy”, “moderation”, and “balance” [26]. DQI is expressed as a percentage: higher percentages translate into a closer affinity towards the national recommendations [27].

Dietary patterns were identified using factor loadings with principal component analysis (PCA). Food groups were initially selected using factor loadings <−0.3 and >0.3, which were defined as items infrequently and frequently consumed, respectively. Rotation was carried out by Varimax orthogonal rotation. From the visual scree plots, a component number of 5 was used as a cut off in determining the most variability within the components. Consumption of each dietary pattern by UC, CD, and controls were then analysed by consumption frequency in tertiles: with the tertile 1 indicating low frequency of consumption. Differences in dietary patterns between IBD and controls adjusting for age, gender, ethnicity, and total energy intake were further analysed by logistic regression analysis. Logistic regression analysis was used as the outcome variable was binary when comparing the dietary data for UC with non-IBD, and CD with non-IBD.

### 2.4. Ethical Considerations

This study was conducted with approval of the Health Research Authority, United Kingdom [REC reference: 20/NE/0118] on 14 May 2020. This study was funded by the St. Mark’s Academic Foundation, London.

## 3. Results

### 3.1. Population Demographics and IBD Characteristics

A total of 286 participants completed the SCGFFQ: 126 non-IBD controls, and 160 IBD cases (114 UC and 46 CD). The median age across the groups was 40 years (IQR = 24) for UC, 34 years (IQR = 29) for CD and 36 years (IQR = 24) for non-IBD control. The control groups included 60/126 females and 76/126 Caucasian individuals.

Table 1 summarises the baseline population demographics and disease demographics for the IBD cohort. Most patients with CD had non-stricturing, non-penetrating terminal ileal disease. There were no patients with upper GI CD, and only one had perianal CD at diagnosis. In the IBD cases, there were 11 current smokers, 26 ex-smokers, and 87 non-smokers.

### 3.2. Macro- and Micro-Nutrient Intakes

Both males and females with CD had the lowest daily total energy intake (kcal/d) compared with UC and controls: 1711 kcal/d (SD 684) and 1910 kcal/d (SD 940) for males and females, respectively, compared with 2575 kcal/d (SD 1226) and 2595 kcal/d (SD1237) for UC and 2393 kcal/d (SD 689) and 2580 kcal/d (SD 1201) for controls (*p*-value < 0.0005).

Total energy consumption (males: 2575; females: 2595 kcal/d) in the UC group was in excess of the UK-recommended DRVs (2500 kcal/d for males and 2000 kcal/d females is 2000 kcal/d) [25], whilst CD patients did not meet the DRVs.

In contrast, the relative percentage of energy derived from protein, carbohydrates, and fats was similar across all three groups (Appendix A).

The mean DQI were 59.0% (SD 18.0) for UC, 46.0% (SD 17.7) for CD, and 63.2% (SD 17.1) for control groups.

Micronutrient analysis showed significantly less consumption of both minerals and vitamins in CD patients compared to UC and controls. In contrast, UC patients had higher consumption of the following micro-nutrients: sodium, chloride, calcium, phosphorous, selenium, iodine, retinol, vitamin D, riboflavin, niacin, vitamin B6, vitamin B12, and panthanoic acid levels (Table 2).

### 3.3. Dietary Patterns

The PCA identified five dietary patterns for the IBD cohort and controls. For each dietary pattern, the frequently and infrequently consumed food items are listed in Table 3.

Pattern 1 was the most varied for food items, whilst patterns 3 and 4 were primarily animal- and fish-based diets, respectively. Dietary pattern 2 was rich primarily in carbohydrates, inclusive of refined sugars, but low in fibre. Dietary pattern 5 consisted primarily of food rich in refined sugars and saturated fats.

Dietary pattern 1 was significantly associated with healthy controls (Spearman’s correlation, *p*: 0.05) (Figure 1). Healthy controls were observed to have a higher frequency of consumption of dietary pattern 3, though this did not meet statistical significance. No statistically significant differences between IBD and controls were observed for the other dietary patterns. There was a trend towards the higher consumption of dietary patterns 2 and 5 in the UC cohort. Dietary pattern 1 was less likely to be consumed by the CD population.

## 4. Discussion

This is the first study to explore dietary patterns, diet quality and macro- and micro-nutrient intake in the months preceding an IBD diagnosis. We found that newly diagnosed CD patients were more likely to have a low-quality diet, with widespread macro- and micro-nutrient deficiencies, and very low diet quality indices.

We noted five dietary patterns within this newly diagnosed IBD cohort. Dietary pattern 2 and 5 consisted of predominantly carbohydrates, predominantly refined sugars; consumption of these patterns was more common in UC, wherein total caloric intake was predominantly driven by the carbohydrate component of their diet (Appendix A). This finding was not replicated in the CD group and may be explained by the overall lower consumption of both macro- and micro-nutrient intake in this group. The lower DQI score seen in the CD cohort compared to controls further supports the significantly reduced nutrition intake. It is noteworthy that dietary pattern 2 is characterised by low fibre intake. Studies in animal models suggest that low-fibre diets promote intestinal microbial dysbiosis and mucosal degradation [28]. However, population studies report conflicting results: a large European cohort study failed to show an association with low fibre intake and development of UC [29] yet, several other studies including the Nurses’ Health Study showed a low risk of CD with high fibre consumption [30,31]. The interplay between CD risk and fibre consumption has also been shown to be dependent on smoking status, and whether this is derived primarily from fruit or vegetable intake. Non-smokers and consumption of cereal-based fibres; and fruit fibre consumption has previously been shown to be associated with lower CD risk [17,29,30].

With respect to micro-nutrient intake, UC patients reported a higher intake of retinol, vitamin D, riboflavin, niacin, vitamin B6, vitamin B12, and panthanoic acid, conforming with a diet rich in animal products, but lower in fruit/vegetable intake. Retinol is a derivative of vitamin A from a carnivorous source, whilst carotenoids represent a plant-based source. Vitamin A has been shown to be protective to intestinal mucosal barrier integrity [32]. Dietary patterns in early studies on IBD identified associations with diets rich in animal products, such as red/processed meats, and specific nutrients such as refined sugars and saturated fats [33]. 

The results of these were often conflicting when looking specifically at the risks of developing UC or CD [34,35,36,37,38,39].

Dietary patterns recently shifted to incorporate ultra-processed foods, and their relationship with IBD was studied [40]. The definition of ultra-processed foods has also evolved from food items that are low in nutrient value, highly processed, rich in saturated fats and refined sugars to the NOVA-4 classification system, which refines the original definition of processed foods by categorising each food item by the degree and extent of processing from its natural state to the finished marketed product [41]. A NOVA classification of 4, labelled as ultra-processed, is assigned to the most industrialised food items, which have been significantly altered from its natural state [42]. For example, unpasteurised milk would be classified as NOVA-1, whilst powdered milk would be considered a NOVA-4 UPF. However, applying the NOVA-4 classification in IBD has shown conflicting results in two prospective cohort analyses of whole population databases; one study reported an association with CD, whilst the other reported an association with both UC and CD [14,43]. The application of the NOVA-4 classification to diet in IBD may be too reductive, as EEN, whilst meeting the criteria for NOVA-4 UPF, is a very effective treatment strategy in Crohn’s disease. In our study, we failed to observe a clear distinction between UPF and non-processed foods correlating with the onset of IBD.

Another measure of dietary components and IBD risk is the inflammatory potential of the diet. The empirical dietary pattern (EDIP) was a recently proposed marker of a pro-inflammatory diet that identified 18 food groups, correlated with three pro-inflammatory serum markers: IL-6, CRP, TNFα [44]. A higher EDIP score correlated with higher pro-inflammatory serum markers. However, higher EDIP scores were not associated with a higher risk of association with UC [44]. In our study, dietary patterns 2, 3, 4, and 5 were more likely to have a high pro-inflammatory EDIP score, driven by the frequently consumed animal and fish protein, and refined dietary sugars within those dietary patterns, and yet we did not observe any statistically significant associations within the IBD cohort. This suggests that the relationship between dietary patterns and IBD onset may be more nuanced than a purely UPF versus non-UPF diet.

We acknowledge some limitations to our study. Dietary research in IBD is fraught with challenges, ranging from the complexity of inflammation biology and its myriad triggers but also by the multiple putative mechanisms by which food components affect the gut and induce symptoms. Our population was small, introducing bias from several known and unknown confounders to the interpretation of the observations. Our CD population was also smaller than the UC group. We had limited data, such as for smoking and weight, which are recognised modulators of the gut microbiome and reflect diet adequacy. We matched controls for age, sex and ethnicity but not for other environmental or lifestyle factors. Unlike previous studies we did not record BMI. The value of BMI in assessing nutritional status, however, has recently been challenged, especially in non-Caucasian cohorts [45], with suggestions that it should be replaced with more accurate measures, such as bone density assessments and bioelectrical impedance analysis [46]. We also did not specifically assess for concomitant irritable bowel syndrome in the IBD population, a syndrome commonly seen in IBD independent of disease activity [47]. With questionnaire studies, recall bias may influence questionnaire completion and susceptibility to the Hawthorne effect, whereby both patients and controls may have overstated (what they perceive to be) healthy food intakes. The study was not designed to examine a temporal relationship between diet and IBD onset, leaving uncertainty as to whether the evident restrictive intake predisposes to CD, or whether the severe CD symptom burden restricted dietary intake.

The limitations of our study and dietary research itself notwithstanding, our study has notable strengths. This is the first study to explore dietary patterns, diet quality, and macro- and micro-nutrient intake in the months preceding an IBD diagnosis. We included a recently diagnosed cohort where dietary intake was assessed by a questionnaire that was validated in different ethnic groups. An additional strength is the prospective nature of our data collection and analysis of diet quality and nutrient intake.

## 5. Conclusions

Newly diagnosed CD demonstrated a low intake of both macro- and micro-nutrients with a very low diet quality index compared to both the control group and the recommended national guidelines. Such a pattern was not observed in UC. The impact of the malnourished state in setting the course of CD early after the diagnosis warrants further study. Nonetheless, this finding lends supports the importance of early dietary intervention following a diagnosis of CD in improving nutritional deficiencies and well-being. In parallel, an animal-based diet rich in sugars and fats was associated with UC (but not CD), consistent with previous reports. Further research into these specific dietary patterns may offer insights into the inflammatory pathways leading to the onset of UC.

Whilst the application of such findings to modulate IBD risk remains elusive, they underscore the value of dietary attention early in the disease course. The impact of early nutrient replacement on the natural history of the disease by altering the activation of inflammatory pathways merits further research. Our observations should serve to inform clinical studies through the identification of nocebo effects and pro-inflammatory food groups, enabling deeper mechanistic insights into the effects of food on the microbiota and the immune system.

The dietary patterns identified here are a proof of concept, and the next phase of the study would be to monitor these patterns in a case–control cohort prospectively, and to further understand the mechanisms behind which the dietary patterns influence IBD by including serum levels of micro- and macro-nutrients, with inflammatory biomarkers and serum and stool metabolomics analysis.

A deeper understanding of dietary-driven inflammatory pathways is within reach following the evolution of artificial intelligence and bioinformatics, whereby various dietary dimensions and dietary measures can be integrated with clinical data to deepen the associative links to disease onset or disease relapse.

## Figures and Tables

**Figure 1 nutrients-17-00958-f001:**
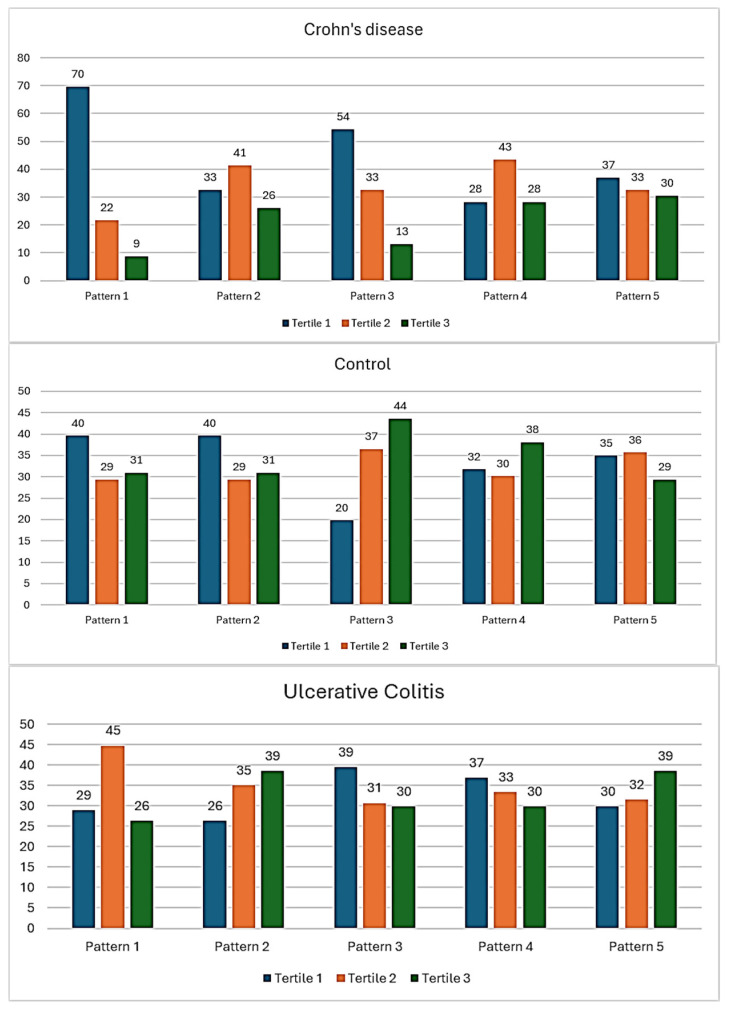
Summary of the consumption of the dietary patterns within UC, CD, and controls. Comparison of the consumption of five dietary patterns between UC, CD, and controls. Count on y-axis refers to percentage frequency. Blue: 1st tertile, Orange: 2nd tertile, Green: 3rd tertile.

**Table 1 nutrients-17-00958-t001:** Patient demographics and IBD characteristics as per the Montreal Classification.

	Ulcerative Colitis *n* = 114 (%)	Crohn’s Disease *n* = 46 (%)	Controls *n* = 126 (%)
**Median age, years** **IQR**	40 24	34 29	40 24
**Gender**			
Female	44 (39)	28 (61)	44 (35)
Male	69 (61)	18 (39)	82 (65)
Not recorded	1 (0.8)	-	
**Ethnicity**			
Caucasian	73 (64)	40 (87)	73 (58)
Non-Caucasian	41 (36)	6 (13)	53 (42)
**Montreal classification** Location Behaviour	E1: 31 (27) E2: 40 (35) E3: 43 (38) - - - -	L1: 19 (41) L2: 15 (33) L3: 12 (26) B1: 33 (72) B2: 11 (24) B3: 2 (4) p: 1 (2)	
**Smoking history** Non-smoker Current smoker Ex-smoker Not stated	44 (39) 11 (10) 25 (21) 34 (30)	43 (93) 0 1 (2) 2 (4)	

**Table 2 nutrients-17-00958-t002:** Mean micronutrient intake for UC, CD, and control. Red highlights indicate an intake higher than controls; blue highlights indicate a lower intake than controls. Significant *p*-values highlighted in bold.

	Ulcerative Colitis *n* = 114		Crohn’s Disease *n* = 46		Control *n* = 126		*p*-Value
**Minerals**							
	**Mean**	**SD**	**Mean**	**SD**	**Mean**	**SD**	
**Sodium mg**	3353.0	1671.4	2383.7	1162.8	3352.3	1531.4	**0.00044**
**Potassium mg**	4133.2	1985.7	2631.6	1279.1	4289.8	1851.0	**<0.0001**
**Calcium mg**	1292.5	611.3	845.8	434.9	1261.7	535.3	**<0.0001**
**Magnesium mg**	378.3	170.5	242.8	120.9	408.1	171.9	**<0.0001**
**Phosphorous mg**	1820.5	856.8	1221.2	615.0	1781.6	674.1	**<0.0001**
**Iron mg**	14.9	6.7	10.1	5.4	16.5	7.0	**<0.0001**
**Copper mg**	1.7	0.8	1.0	0.5	2.2	4.5	**<0.0001**
**Zinc mg**	12.1	5.6	8.7	4.9	12.8	6.5	**0.00003**
**Chloride mg**	5062.0	2469.1	3614.8	1771.8	5011.5	2268.5	**0.00073**
**Manganese mg**	3.9	1.8	2.6	1.4	4.6	2.2	**<0.0001**
**Selenium ug**	73.1	59.9	51.6	37.7	70.2	34.2	**0.00041**
**Iodine ug**	299.4	224.1	189.3	117.1	240.7	103.1	**0.00031**
**Vitamins**							
	**Mean**	**SD**	**Mean**	**SD**	**Mean**	**SD**	
**Retinol ug**	525.7	419.0	343.7	266.4	431.7	286.2	**0.00244**
**Carotenoids ug**	4699.4	3336.7	2346.9	2000.5	7788.4	5675.7	**<0.0001**
**Vitamin D ug**	5.2	6.3	3.8	3.4	4.6	3.5	0.18781
**Vitamin E mg**	13.6	7.5	8.3	3.9	14.7	7.3	**<0.0001**
**Thiamine mg**	1.9	0.9	1.3	0.7	2.0	0.8	**0.00002**
**Riboflavin mg**	2.3	1.2	1.6	1.0	2.1	0.9	**0.00005**
**Niacin mg**	23.6	13.5	17.7	10.9	22.3	8.5	**0.00179**
**Vitamin B6 mg**	2.6	1.3	1.8	1.0	2.5	1.0	**0.00003**
**Vitamin B12 ug**	8.2	8.1	5.9	4.9	6.4	4.2	**0.03772**
**Folic acid ug**	321.9	140.8	202.0	110.2	369.5	167.2	**<0.0001**
**Panthanoic acid mg**	6.9	3.2	4.7	2.5	6.6	2.6	**<0.0001**
**Biotin ug**	50.8	25.9	32.9	20.4	51.7	21.9	**<0.0001**
**Vitamin C mg**	130.9	78.6	66.1	46.4	164.7	110.2	**<0.0001**

**Table 3 nutrients-17-00958-t003:** Table showing the frequently and infrequently consumed food items for the five key dietary patterns, with a descriptive summary of the main nutrient component of the pattern in the third column.

	Frequently Consumed	Infrequently Consumed	Characteristics
Pattern 1	Fruit (including fresh juice) Vegetables including salads Puddings, cakes and ice cream Sauces and condiments Cheese, full fat Beans/pulses Fish, including oily, white, processed Eggs Nuts Chocolate and sweets Biscuits Pizza/quiche Soup Pasta and noodles Potatoes		Variety of food items Consists of oily fish
Pattern 2	Biscuits, sweetened Spreads, sweetened Breakfast cereals Chocolate and sweets. Sauces and condiments Crisps and savoury snacks Cheese, full fat Processed fruit drinks	Fruit Yoghurt, low fat/calorie Beans/pulses Soup Tea Water Vegetables	Rich in carbohydrates and refined sugars Low fibre
Pattern 3	Meat including red, white and processed Fish including oily and white Potatoes	Biscuits, sweetened Crisps and savoury snacks Breakfast cereals unsweetened Spreads, sweetened Beans/pulses Meat substitute Rice, brown Sauces and condiments	Primarily animal- and fish-based diet
Pattern 4	Fish, including oily, white, processed Brown rice Crisps and savoury snacks Breakfast cereals, unsweetened Spreads, sweetened Sauces and condiments		Fish-derived protein sources with carbohydrate
Pattern 5	Fruit juice, processed and fizzy Fizzy diet drink Potatoes Meat substitute Puddings and ice cream	White rice Cheese, full fat Bread, high fat Eggs	Refined sugar and high saturated fat

## Data Availability

The data will be available on request.

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
