# Peer review of "A Low Dietary Quality Index in a Newly Diagnosed Inflammatory Bowel Disease Cohort: Results from a Case—Control Study"

_nutrients, 2025, doi:10.3390/nu17060958_

Round 1
Reviewer 1 Report
Comments and Suggestions for Authors
A low dietary quality index in a newly diagnosed inflammatory bowel disease cohort: results from a case control study
The manuscript presents an important and insightful study on the dietary patterns and nutrient intake of patients with newly diagnosed inflammatory bowel disease (IBD). The study's focus on diet quality and the relationship between dietary intake and IBD progression is commendable, as it adds valuable evidence to an area with significant clinical relevance. The methodology, including the use of validated dietary questionnaires and the principal component analysis (PCA) for identifying dietary patterns, is sound and well-executed.
Suggestions for Improvement:
- The abstract provides a good summary of the background and methods, but the results section could be more explicit. For example, including exact figures or trends regarding the dietary intake differences between UC, CD, and controls would provide more immediate clarity on the study's findings.
- The control group is described as including non-IBD patients, healthcare workers, and university students. It would be useful to clarify whether these controls are matched for diet and lifestyle factors, as they may differ significantly from the general population in ways that could affect dietary patterns.
- The study accounts for ethnicity in the analysis but could delve deeper into how different ethnicities might affect the dietary patterns or nutrient deficiencies observed in IBD patients. Providing more specific data on the ethnic breakdown could enhance the study's validity.
- The PCA identified five dietary patterns, but the analysis could be expanded to include a discussion on why certain dietary patterns were more prevalent in IBD patients and what this might imply for disease management. Also, a more detailed breakdown of the food items that define each pattern would enhance understanding.
- While the study highlights deficiencies in CD patients, it would be helpful to discuss the clinical implications of these deficiencies more thoroughly. How might these nutritional shortfalls impact disease progression or treatment outcomes?
- The statistical methods section mentions using the Kruskal-Wallis test and logistic regression but does not explain why these methods were chosen. Further justification for the statistical approach and its relevance to the study design would provide more rigor.
- While the manuscript emphasizes the importance of early dietary intervention, it would be valuable to expand on how specific dietary changes could be implemented in clinical practice. Are there any existing intervention strategies for newly diagnosed IBD patients that could be incorporated?
- The discussion acknowledges some conflicting results in the literature concerning diet and IBD, which is good. However, the authors should more explicitly address how their findings relate to or challenge existing studies. This could provide more context for the study's contributions to the field.
Author Response
Comment 1
The abstract provides a good summary of the background and methods, but the results section could be more explicit. For example, including exact figures or trends regarding the dietary intake differences between UC, CD, and controls would provide more immediate clarity on the study's findings.
Response 1
Thank you raising this point. We have added the following line in the abstract results section.
‘There was a trend towards higher consumption of dietary pattern 2 (rich in carbohydrates, refined sugar and low fibre) and 5 (refined sugar and saturated fat) in the UC cohort. Dietary pattern 1 (variety of food items and oily fish) was less likely to be consumed by the CD population.’
Comment 2
The control group is described as including non-IBD patients, healthcare workers, and university students. It would be useful to clarify whether these controls are matched for diet and lifestyle factors, as they may differ significantly from the general population in ways that could affect dietary patterns.
Response 2
We agree the control population is not generalisable to the IBD population with regard to all lifestyle factors. We matched controls for age, sex and ethnicity. We have acknowledged this as a limitation in the discussion at page 23, line 318.
Comment 3
The study accounts for ethnicity in the analysis but could delve deeper into how different ethnicities might affect the dietary patterns or nutrient deficiencies observed in IBD patients. Providing more specific data on the ethnic breakdown could enhance the study's validity.
Response 3
We agree that ethnicity is a very important factor that might affect dietary patterns. However as this was not the primary aim of the study the numbers recruited in Non-Caucasian groups were inadequate for deeper analysis when split into individual ethnic groups.
Comment 4
The PCA identified five dietary patterns, but the analysis could be expanded to include a discussion on why certain dietary patterns were more prevalent in IBD patients and what this might imply for disease management. Also, a more detailed breakdown of the food items that define each pattern would enhance understanding.
Response 4
Thank you for your comment. This study focused on the possible dietary patterns that may be associated with the onset of IBD. The dietary patterns identified here are a proof of concept, and the next phase of the study would be to ideally monitor these patterns in a case-control cohort prospectively to further understand the mechanisms behind which the dietary patterns influence IBD by including serum levels of micro-and macro- nutrients, with inflammatory biomarkers, and serum and stool metabolomics analysis. It is therefore beyond the scope of this study to correctly prescribe a dietary pattern to manage IBD.
The breakdown of food for each dietary pattern is given in Table 3.
Comment 5
While the study highlights deficiencies in CD patients, it would be helpful to discuss the clinical implications of these deficiencies more thoroughly. How might these nutritional shortfalls impact disease progression or treatment outcomes?
We entirely agree longitudinal follow up of these patients to assess the impact of these deficiencies on the clinical outcomes would be important but unfortunately beyond the scope of this study as we do not have clinical outcome data. We have concluded that early dietary intervention should be considered in newly diagnosed patients with Crohn’s.
Comment 6
The statistical methods section mentions using the Kruskal-Wallis test and logistic regression but does not explain why these methods were chosen. Further justification for the statistical approach and its relevance to the study design would provide more rigor.
Response 6
Thank you for your comment.
The Kruskal-Wallis test was used to allow comparison of non-IBD, UC and Crohn's data, which was not normally distributed, and was non-parametric.
Logistic regression analysis was used as the outcome variable was binary when comparing the dietary data for UC with non-IBD, and CD with non-IBD.
This has been added to the methods section at the following lines respectively; page 10 line 202 and page 11 line 222.
Comment 7
While the manuscript emphasises the importance of early dietary intervention, it would be valuable to expand on how specific dietary changes could be implemented in clinical practice. Are there any existing intervention strategies for newly diagnosed IBD patients that could be incorporated?
Response 7
Thank you for your comment. We would like to recommend a specific intervention strategy however the sample size of Crohn’s patients is small with mainly non-stricturing and non-penetrating disease. The sample population would limit the generalisability of any recommendations. An intervention strategy would be tailored to the specific requirements of the individual. A detailed assessment including micronutrient analysis should be carried in patients to identify specific deficiencies that may need to be corrected.
Comment 8
The discussion acknowledges some conflicting results in the literature concerning diet and IBD, which is good. However, the authors should more explicitly address how their findings relate to or challenge existing studies. This could provide more context for the study's contributions to the field.
Response 8
The use of dietary pattern analysis is novel in comparison to previous studies which also makes direct comparison with other studies difficult. We propose further studies should apply dietary pattern analysis which is more representative and clinically applicable rather than macro- and micronutrient intake. This has been added to the conclusion at line page 25 line 417-420.
Reviewer 2 Report
Comments and Suggestions for Authors
In my point of view, the study conducted by Misra et al. requires the following modifications.
In the abstract, it would be useful if some directions for further studies could be indicated.
Keywords are not indicated.
More worldwide studies on inflammatory bowel diseases, particularly those ones about Crohn’s disease and ulcerative colitis should be cited in the Introduction.
Regarding the ethical considerations, the approval date should be indicated.
How did the authors find the sample size (286) representative of the study population? This needs to be justified. The performing of a power analysis would be recommendable.
Tables have to be properly formatted according to the journal’s guidelines.
Elaborate on your conclusions by indicating more directions for further investigations.
Author Response:
In my point of view, the study conducted by Misra et al. requires the following modifications.
Comment 1
In the abstract, it would be useful if some directions for further studies could be indicated.
Response 1
Thank you for your comment. We have added the following sentence to the abstract conclusion.
‘The dietary patterns identified here are a proof of concept, and the next phase of the study would be to ideally have been to monitor these patterns to a case-control cohort prospectively, and to further understand the mechanisms behind which the dietary patterns influence IBD.’
Comment 2
Keywords are not indicated.
Response 2
Thank you. This have been removed from the abstract page.
Comment 2
More worldwide studies on inflammatory bowel diseases, particularly those ones about Crohn’s disease and ulcerative colitis should be cited in the Introduction.
Response 2
We acknowledge this and have added the following two studies to the introduction at page 7 line 139-142.
Inflammatory bowel disease has become a global disease with accelerating incidence in newly industrialised countries whose societies have become more westernised. Although incidence is stabilising in western countries, burden remains high as prevalence surpasses 0·3%. (1) For example in South Asia there has been a rapid increase in IBD incidence and prevalence. (2)
- Ng SC, Shi HY, Hamidi N, Underwood FE, Tang W, Benchimol EI, et al. Worldwide incidence and prevalence of inflammatory bowel disease in the 21st century: a systematic review of population-based studies. Lancet (London, England) [Internet]. 2017 Oct 13 [cited 2017 Dec 5];0(0). Available from: http://www.ncbi.nlm.nih.gov/pubmed/29050646
- Shenoy S, Jena A, Levinson C, Sharma V, Deepak P, Aswani-Omprakash T, et al. Inflammatory bowel disease in south Asia: a scoping review. Lancet Gastroenterol Hepatol [Internet]. 2025;10(3):259–74. Available from: http://dx.doi.org/10.1016/S2468-1253(24)00341-8
Comment 3
Regarding the ethical considerations, the approval date should be indicated.
Response 3
Thank you for raising this point. The approval date of 14th May 2020 has been added at page 11 line 230.
Comment 4
How did the authors find the sample size (286) representative of the study population? This needs to be justified. The performing of a power analysis would be recommendable.
Response 4
Many thanks for your comment. The research was conducted as part of an epidemiological study to examine the incidence of newly diagnosed IBD in 8 urban centres. It was not possible to carry out a power calculation for this inception cohort study as the number of questionnaires would depend on the number of incident cases. Of the newly diagnosed patients over 1 year throughout the UK, 286 patients completed the questionnaire. We acknowledge a dedicated power calculation would add validity and acknowledge this as a limitation.
Comment 5
Tables have to be properly formatted according to the journal’s guidelines.
Response 5
Thank you for your comment. According to the journal guidelines the minimum size font is 8. We have ensured all font sizes are greater than 8. We are happy to make any adjustments to the tables as required.
Comment 6
Elaborate on your conclusions by indicating more directions for further investigations.
Response 6
The dietary patterns identified here are a proof of concept, and the next phase of the study would be to ideally have been to monitor these patterns to a case-control cohort prospectively, and to further understand the mechanisms behind which the dietary patterns influence IBD by including serum levels of micro-and macro- nutrients, with inflammatory biomarkers, and serum and stool metabolomics analysis. This has been added to the conclusions at page 25 line 420-425.